# A Heterogeneity-Aware Replacement Policy for the Partitioned Cache on Asymmetric Multi-Core Architectures

**DOI:** 10.3390/mi13112014

**Published:** 2022-11-18

**Authors:** Juan Fang, Han Kong, Huijing Yang, Yixiang Xu, Min Cai

**Affiliations:** Faculty of Information Technology, Beijing University of Technology, Beijing 100124, China

**Keywords:** asymmetric multi-core, last-level cache, replacement policy, heterogeneity-aware

## Abstract

In an asymmetric multi-core architecture, multiple heterogeneous cores share the last-level cache (LLC). Due to the different memory access requirements among heterogeneous cores, the LLC competition is more intense. In the current work, we propose a heterogeneity-aware replacement policy for the partitioned cache (HAPC), which reduces the mutual interference between cores through cache partitioning, and tracks the shared reuse state of each cache block within the partition at runtime to guide the replacement policy to keep cache blocks shared by multiple cores in multithreaded programs. In the process of updating the reuse state, considering the difference of memory accesses to LLC by heterogeneous cores, the cache replacement policy tends to keep cache blocks required by big cores, to better improve the LLC access efficiency of big cores. Compared with LRU and the SRCP, which are the state-of-the-art cache replacement algorithms, the performance of big cores can be significantly improved by HAPC when running multithreaded programs, while the impact on little cores is almost negligible, thus improving the overall performance of the system.

## 1. Introduction

The power density problem in processors is the main reason for the arrival of the multi-core era. Further increasing the clock frequency by increasing the density of transistors will lead to difficult chip heat dissipation problems, which makes it very difficult to improve the performance of a single processor. It is almost impossible to improve the overall performance of a computer system by increasing the performance of a single thread, but only through parallelization [1,2]. Herein, multi-core processes are increasingly used in computer systems.

Modern computer systems are more diverse than ever, ranging in size from handheld embedded machines to large cloud computing centers. In terms of utilization, those computer systems exhibit intermittent inactivity, and peak resource demandsa huge difference [3,4,5,6]. To meet the growing demand for computer systems, the future computer system needs to have higher scalability, and the heterogeneous multi-core system is the key technology. In particular, a single ISA asymmetric multi-core processor (AMP), also known as a heterogeneous multi-core processor (HMP), uses heterogeneity as a first principle. The processor uses different types of cores, and each type of core has a different microarchitecture design [7,8]. For example, some cores in the processor are designed with sequential pipelines, and some are designed out of order; some cores are in MT (multithread) mode, and some are in ST (single-thread) mode; some cores support large cache structures, and some cores support small cache structures. It can be optimized for power/performance or different application domains, or can take advantage of instruction-level parallelism (ILP), thread-level parallelism (TLP), or memory-level parallelism (MLP). Therefore, asymmetric multi-core processors are easier to adapt to diversity and are expected to play a role in a wide range of usage scenarios [3]. The processors share the same instruction set architecture and it can be decided at runtime by the operating system which task/thread is mapped to which core. As a result, asymmetric multi-core processors can both accommodate the diversity of core microarchitecture designs and leverage load balancing for fine-grained control over performance and power consumption.

However, in multi-core systems, including symmetric multi-core processors (SMP) and AMP systems, concurrently running threads compete with each other for shared resources. Threads may be slower than when running individually and monopolizing shared resources such as shared cache, and memory bandwidth. In the AMP system, the problem of contention for shared resources is more serious [9,10]. Since the high-performance cores cannot respond faster from the shared resources, their performance cannot be effectively improved. The effectiveness of the AMP system is not fully utilized, and the use of high-performance cores is reduced. Therefore, if contention for shared resources in an AMP system is not effectively managed, it may degrade the performance of individual threads (especially those running on high-performance cores) and the overall system [11,12,13]. Furthermore, in the AMP system, the memory access streams of different threads are interleaved and interfere with each other in the shared last-level cache (LLC). Interthread interference destroys the original spatial locality of a single thread, thus seriously degrading the system performance. As the number of cores on a chip in an AMP continues to increase, the contention and interference for limited shared memory resources become more serious, and effective management of shared LLC resources is extremely important for AMP systems. Cache partitioning reduces intercore interference by dividing the shared LLC among different cores. However, the current cache partitioning strategies proposed for SMP are based on the assumption that all cores have the same performance [14,15]. In AMP, it needs to be adjusted according to the performance between different cores in order to take advantage of the different advantages of asymmetric performance cores.

To maximize the efficiency of the AMP system, we propose a heterogeneous-aware partitioned cache replacement policy framework (HAPC). HAPC considers cache block reuse during cache replacement and avoids the need for core interference. The key idea of HAPC is to analyze the memory access characteristics of cores and threads at runtime, and guide the replacement policy by sensing the LLC requirements of heterogeneous cores. In general, this paper makes the following contributions:We propose a heterogeneous-aware partitioned cache replacement policy, which reduces intercore interference and improves the efficiency of data usage in partitioned LLC in an asymmetric multi-core architecture.We design a reuse count table (RCT) for the historical reuse information of each cache block in LLC, and update the value inthe RCT according to the memory access characteristics of big cores and little cores, which can be further used in cache replacement decisions.Finally, we evaluate HAPC in detail with PARSEC 3.0 using the gem5 simulator. On average, HAPC improves the performance of big cores by 4.57% and 2.44% over LRU and SRCP, respectively. HPAC can provide effective performance improvement relative to the traditional replacement policies in various workloads and system configurations.

The second part of this paper presents the basic background and related work. Section 3 explains our HAPC framework. The fourth part introduces the experimental method and analyzes the experimental results. Finally, Section 5 concludes the paper.

## 2. Related Work

### 2.1. Asymmetric Multi-Core Architecture

As the heterogeneity of AMP increases, the complexity of AMP management rises exponentially. For this reason, traditional SMP optimization techniques may not work well in AMP, so new techniques are needed to manage AMP. In recent years, many related technologies have been proposed to meet the needs of AMP management [16].

Zhao et al. [17] divides asymmetric multi-core processors into performance asymmetric multi-core processors, functional asymmetric multi-core processors, and dynamic multi-core processors, based on the differences in microarchitecture of different cores, from the aspects of instruction-level architecture, pipeline differences, and cache system differences. Among those processors, the performance asymmetric multi-core processor generally integrates a big core with high performance and high power consumption, and a little core with low power consumption and low performance in one processor chip to provide services for applications with different requirements. The ARM big.LITTLE big and little core architecture is the most successful commercial asymmetric multi-core architecture design. It integrates high-performance processors and low-power processors through SoC design, such as Samsung Exynos 9 series processors [18]. The processor has two clusters: a big core cluster and a little core cluster. The big core cluster uses the Cortex A15 with higher performance, the little core cluster uses the Cortex A7 with lower power consumption, and the L2 Cache is shared within the cluster. This heterogeneous design of high performance combined with low-power CPU cores can provide high-performance processing power at significantly lower average power consumption.

The optimization goal under the asymmetric multi-core architecture is that the collaborative computing cores can work better when running different programs, so as to improve the overall performance. To fully realize the potential of AMP, it is necessary to solve the challenges of complex architecture design, including task division, task mapping, data communication, data parallelism, etc., between heterogeneous cores [19,20,21]. Liu et al. [22] analyzed that the complexity of scheduling on AMP grows exponentially with the increase of core types and the number of applications. Jia et al. [23] proposed a dynamic resource partitioning method for a single ISA heterogeneous multi-core. This method divides shared resources according to the requirements of two threads for shared resources and the performance of the running core, which is very effective in improving the throughput and fairness of a single ISA asymmetric multi-core system.

### 2.2. Shared Last-Level Cache Management Policies

A shared last-level cache means that multiple threads can share some data, reduce communication delay, reduce redundant backup of data, and improve cache space utilization. However, contention between threads for limited cache space will also lead to an increase in the cache miss rate, which affects the throughput and fairness of the system. In order to reduce the impact of competing for cache space between threads, the optimal cache block can be reserved by designing a reasonable cache replacement policy. Another intuitive idea is to partition the cache and avoid interthread interference by explicitly allocating cache space to each core.

The cache partitioning technique divides and distributes the entire LLC space to each core, and ensures that each core fully occupies the allocated area, thereby intentionally eliminating interference between applications and alleviating cache space imbalance between applications. Cache partitioning combines the isolation of private caches and the high capacity of shared caches, and is suitable for a wide range of application scenarios. Qureshi et al. proposed a utility-based cache partitioning (UCP) [24] method to minimize the total number of misses incurred by all applications in the workload on the shared last-level cache. Huang et al. [25] proposed a low-power shared cache partitioning method with combined linear and exponential curve fitting considering the characteristics between cache miss rate and cache size. Pons et al. [26] analyzed LLC behaviors detrimental to cache performance, data reuse, and cache occupancy, and based on these behaviors, a critical phase-aware partitioning approach (CPA) was proposed, which achieves IPC-preserving by efficiently utilizing LLC space while reducing turnaround time.

Cache replacement policies mainly use the principle of locality to reduce intra-application interference on single-core systems. Besides traditional replacement policies (such as FIFO, MRU, LFU, and LRU), many high-performance replacement policies have been proposed to apply to shared LLC [27,28,29]. Ref. [29] learns from Belady’s optimal solution for past references to predict the caching behavior of future references, but it is not designed for asymmetric multi-core architecture. Those policies consider the workload characteristics of individual applications and dynamically select a policy suitable for each application to reduce the intra-application interference of each application, but the replacement policy cannot fundamentally eliminate the interapplication interference on the shared LLC [30]. A sharing and reuse-aware partition cache replacement policy [31] is proposed to track all accesses to cache blocks in the local core and global check partitions for partition cache maintenance, which can be further used for cache replacement decisions to prevent the eviction of cache blocks shared by multithreaded programs across multiple cores, minimizing intercore interference. Refs. [32,33] are priority-aware scheduling that propose fine-grained application characterizations to improve the performance (IPC) of prioritized applications executing under shared-resource contention. Those methods focus on minimizing contention on both the main-memory bandwidth and the LLC by monitoring the pressure that each application inflicts on these resources. Ref. [34] find that smart cache replacement reduces the burden on software to provide intelligent scheduling decisions. Herein, these priority-aware scheduling and HAPC combined can together improve performance.

As mentioned above, effective cache partitioning and cache replacement policies are the keys to improving the efficiency of shared last-level cache usage. However, the last-level cache management technology for multithreaded programs under asymmetric multi-core architecture has not been studied. In this paper, we propose a heterogeneous-aware partition cache replacement policy, which can sense the difference in memory access characteristics of heterogeneous cores and the reusability of cache blocks to meet the needs of heterogeneous cores for LLC.

## 3. Heterogeneity-Aware Replacement Policy for the Partitioned Cache

In this paper, we design a heterogeneity-aware replacement policy (HAPC) for the partitioned LLC in asymmetric multi-core processors. We first briefly describe the data reuse for multithread programs and the interference of memory access in asymmetric multi-core processors, which are closely related to LLC utilization. Then, we describe the implementation steps of HAPC in detail. By tracking the reuse status of cache blocks and distinguishing the memory accesses between big cores and little cores, HAPC can reduce the interference between cores and improve system performance.

### 3.1. Data Reuse for Multithreaded Programs

The advantage of a multi-core system is that multiple cores can run programs in parallel, including multiprocess programs and multithreaded programs. The main difference between multiprocess and multithreaded applications is that data between processes are independent of each other, and a cache block can only be accessed by one core. There are shared data between multiple threads of the same process, and memory access requests from multiple cores may hit the same cache block. Data in a multithreaded application can be private or shared by different threads. If different threads share the same cache block, it is called constructive sharing. If one thread evicts the cache block used by another thread during cache replacement, it is called destructive sharing [31].

In the cache partitioning scheme in this paper, this work adopts a static partitioning method based on way partitioning, which divides each way of the shared LLC equally among all cores. This approach is coarse-grained and easy to implement, but also wastes LLC space if underutilized. Cache partitioning can effectively avoid interference between cores in multiprocess applications because there is only constructive sharing in multiprocess programs. However, for multithreaded programs, cache partitioning can only solve the intercore interference caused by constructive sharing, and cannot effectively deal with destructive sharing. This is because the replacement policy of the cache partition only considers the impact on the LRU stack when the memory access request of the core hits, and may evict the cache blocks shared by multiple cores in the partition. Data sharing becomes a problem that needs to be solved efficiently because each thread executing on a different processor will have a copy of the same data or instructions. Therefore, it is necessary to record the sharing and reuse information of cache blocks in the partitioned cache in response to the above problems, and then guide the replacement policy to improve the constructive sharing across threads and reduce the interference between cores.

In order to retain the cache blocks shared by multiple cores at the same time in the cache partition, we maintain a reuse count table (RCT) for LLC, which is used to record the historical reuse information of each cache block. During the operation of the system, the value in the RCT is updated according to the memory access request of each core, and the selection of the evicted cache block is guided when the cache replacement policy occurs. Assuming that during the running of a multithreaded program, shared data is loaded by a core into a cache block within a cache partition, then the core is called the local core of the cache block, and other cores are called the shared core of the cache block. The RCT of a cache block contains two items, which are used as indicators of the local reuse characteristics and shared reuse characteristics of the cache line:LC (local count): LC is used to record the reuse count caused by the local core fetch request hitting the cache block.SC (share count): SC is used to record the reuse count caused by the shared core memory fetch request hitting the cache block.

Initially, the LC and SC counters in the RCT are set to 0. The cache controller updates the content of the RCT. When a cache hit occurs, the cache controller compares the source of the memory access request with the cache block ownership to determine whether it is LC reuse or SC reuse. When a cache miss occurs, the controller uses the reuse information recorded in the RCT to select an eviction block for the replacement policy.

### 3.2. Memory Access Disturbance in Asymmetric Multi-Core Processor

When the core accesses the shared LLC, it will be mapped to the corresponding cache line according to the request address, and regardless of the cache partition mapped to any core, it can be accessed to complete a cache hit. Once the corresponding cache block cannot be found, a cache miss occurs, and a cache replacement policy needs to be implemented to find the cache block in the core cache partition for eviction and load a new cache block into the LLC. The LRU replacement policy only considers the recent access information of the data block, regardless of the access frequency of the data block and the source of the access request. LRU cannot distinguish the shared cache blocks of multiple cores, and cannot classify the reuse information of the cache blocks accessed by the heterogeneous cores. When a multithreaded program is running with a working set larger than the cache capacity, the cache will thrash, which will cause system performance to degrade.

Generally speaking, in the performance asymmetric multi-core architecture, the cores are divided into two types: big cores and little cores. The two types of cores design are positioned differently. The big core has higher performance, and the little core has poor performance but has higher memory access latency tolerance and lower power consumption. This heterogeneous design of high-performance combined with low-power CPU cores can provide high-performance processing capabilities with significantly lower average power consumption, but there are great differences in memory access characteristics between different types of cores. In order to more intuitively show the difference in memory access between big and little cores, we counted the number of LLC accesses of big and little cores when running parsec multithreaded applications, as shown in Figure 1.

It can be clearly seen from the figure that the number of LLC accesses of the big core is significantly higher than the number of LLC accesses of the little core. This is because the frequency and pipeline depth of the big core are higher than those of the little core. As a result, the memory access requirements of different performance cores are different, and the impact of cache access on other cores in the system is also different. Therefore, when managing LLC, it is necessary to take into account the differences between asymmetric cores to make adjustments. When the memory access request of big cores increases, the cache blocks required by high-performance cores should be reserved as much as possible to ensure that the high-performance core can give full play to the advantages of its microarchitecture design, and improve the overall performance of the system.

### 3.3. Heterogeneous-Aware Partition Cache Replacement Policy

In the heterogeneous-aware partition cache replacement policy (HAPC), the reuse counting table (RCT) tracks the reuse status of cache blocks according to the access frequency of cache blocks. If the access frequency is high, the cache block reuse count is increased, and the cache block has a higher priority to be kept in the LLC; otherwise, the cache block has a lower priority. For the replacement policy, the cache block that has not been accessed relatively recently is selected from the lower priority blocks for eviction. The RCT further classifies the sources of memory access requests, and distinguishes memory requests from the local core and the shared core. The LC counter in the RCT is used to record the reuse of the cache block by the local core. The larger the LC counter, the more times the cache block is reused within the core and has a better locality. The SC counter in the RCT is used to record the reuse of the cache block by the shared core, and marks the cache block shared by multiple cores. If the SC value of the cache block is high, it means that the cache block has high shared reuse characteristics.

Figure 2 shows a high-level overview of the proposed heterogeneous aware partition cache replacement policy. HAPC uses a reuse count table (RCT). An RCT is maintained to keep track of the accesses made to each cache block for every partition and find a replacement block according to the values contained in it. The RCT contains LC and SC, as mentioned in Section 3.1.

The LC and SC of the RCT counter are initially set to 0. The cache controller updates the RCT value according to the process shown in Algorithm 1 when processing the LLC memory access request, and guides the cache replacement policy according to the change of the RCT value. When a cache hit occurs, the cache controller judges the source of the memory access request, differentiates between big and little cores, and controls the LC and SC counters to increase different reuse weights. On a cache miss, the value of the counter present in the RCT is used to find the evicted cache block. At the same time, the LC and SC counter values of the remaining blocks in the cache partition will be decremented by 1, ensuring that the priority of the cache block that will not be accessed again will decrease over time, and will eventually be selected and expelled by the replacement policy; otherwise, it may be long-term retention, causing cache pollution. For a newly inserted cache block, its LC is set to the average value of the LC values of other cache blocks except the evicted cache block in the cache partition.
**Algorithm 1:** RCT update. **Input**: The memory request issued by the core Req; Mapping of LLC ways and     divided cores (Map[x] = c represents that the X_*th*_ way of LLC is divided to      the core c); RCT table, representing reuse information for each cache line;     Reuse weights for different cores (weight_big, weight_little). **Output**: Updated RCT table**1** **if**
*Req hits cache line*
**then****2**  **if**
*core* ∈ *big cores*
**then****3**    LC = LC + weight_big;**4**    SC = SC + weight_big;**5**  **end****6**  **if**
*core* ∈ *big cores*
**then****7**    LC = LC + weight_little;**8**    SC = SC + weight_little;**9**  **end****10** **end****11** **if**
*Req cache misses*
**then**  /* Select the eviction block in the cache partition and execute        the replacement policy                                       */**12**  **for**
*cache line* ∈ *core*
**do****13**        victim = LRU(min(LC) and min(SC));**14**  **end**  /* Insert new data into the cache line and update the        corresponding RCT table                                      */**15**  **for**
*cache line* ∈ *core*
**do****16**        LC = average(LC ∈ core);**17**        SC = 0;**18**  **end**  /* Update other cache blocks in the cache partition            */**19**  **for**
*cache line* ∈ *core*
**do****20**        LC = LC − 1;**21**        SC = SC − 1;**22**  **end****23** **end**

In the heterogeneous-aware partition cache replacement policy, the maintenance of the reuse count table needs to be aware of the memory access behavior characteristics of the heterogeneous cores under the asymmetric multi-core architecture. During program execution, one must monitor the performance indicators of big cores, and dynamically adjust the RCT reuse weights (weight_big, and weight_little) to ensure that cache blocks reused by big cores have higher priority. The larger the value of weight_big is relative to weight_little, the larger the increase in the reuse count in the RCT table when the big core access hits, and the cache block is not easy to evict in the replacement policy. The index of performance judgment is based on the memory access hit rate of the big core. During runtime, the running status of the application is regularly monitored. After a fixed tick interval, the memory access hit rate of the big core is calculated, and then the reuse weight is reset. The specific process is as follows:(1)Set weight_big and weight_little to 1.(2)Calculate the hit rate of the big cores in this interval, and increase the weight_big of the next interval by 1 until the weight_big increases to the threshold (the number of big cores).(3)Calculate the hit rate of the big core in the next interval and compare it with the hit rate of this interval. If the hit rate increases, go to (2); otherwise, go to (1).

## 4. Experiments

### 4.1. Experimental Setup

The experiment uses the gem5 simulator [35] to simulate the ARM big.LITTLE architecture. We use the gem5 full-system simulator to evaluate the heterogeneity-aware replacement policy for the partitioned cache that we proposed. The Ruby memory subsystem in gem5 implements a detailed simulation model, which provides a variety of replacement policies and consistent protocol implementations. The topology structure is built, and the memory access request is forwarded and processed in the DMA and cache controller configuration. The experiment simulates the DIE model of 2 big cores + 2 little cores as an asymmetric multi-core environment, and uses the full system mode to simulate. Both the big cores and the little cores use the ARM O3 CPU type, and the core microarchitecture configuration is shown in Table 1. Each core has its own L1 cache and L2 cache, and the L3 cache is shared between two DIEs. The overall structure of the system is shown in Figure 3.

We evaluate HAPC on the parsec 3.0 [36] benchmark suite, as the benchmark suit has many multithreaded applications. The parsec hook functions also define a region of interest (ROI) for each benchmark. This code region is the part of the benchmark which performs the “interesting” computations. We take the average of all ROIs’ results in each benchmark as the final experimental result. We compare HAPC with the existing cache replacement policy LRU and the sharing and reuse-aware cache replacement policy (SRCP).

A way to core mapping table is established for LLC in gem5 to implement static partitioning based on way partition. We create an RCT table for the LLC to record the reuse status of each cache line, triggering an update of the RCT table entry when a memory access request hits a cache line. When the memory access request misses, the cache controller first filters the cache lines with low reuse characteristics from the RCT table according to the reuse information, and then sends them to the cache replacement policy for LRU to find the eviction block.

### 4.2. Results

We first evaluate the performance of the heterogeneous-aware partitioned cache replacement policy, comparing it with the traditional cache replacement policy LRU, and the sharing and reuse-aware cache replacement policy (SRCP). Figure 4 and Figure 5 show the performance improvement on average for big cores and little cores, respectively. The heterogeneous-aware partition cache replacement policy provides the best performance on average among all parsec sim_small benchmarks. Compared with LRU, HAPC improves the performance of big cores by 4.57% on average, while the performance of little cores decreases by only 0.04% on average, and the decline has little effect on the overall performance. Compared with SRCP, the big core performance of HAPC is improved by an average of 2.44%. Applications such as feeret and streamcluster, which are interthread communication intensive, show more significant improvement in performance than LRU and SRCP approaches. Fluidanimate and swaption are memory nonintensive applications; threads in those applications run in parallel and do not coordinate much, so HPAC shows negligible performance improvement compared to LRU. Applications such as blackscholes and canneals are benefited by HAPCdue to decreased intercore interference in an asymmetric multi-core architecture.

We believe that this is because the LRU and SRCP policies lack awareness of the memory access behavior of heterogeneous cores and do not share resource tilt for big cores. HAPC dynamically adjusts the growth weight of RCT at runtime to ensure that the reuse of big cores has a higher weight, which improves the performance of big cores in an asymmetric multi-core architecture. As a result, HAPC improves the overall performance of asymmetric multi-core systems.

Figure 6 shows the improvement of LLC hit rate of big cores under each configuration. Compared with LRU, HAPC improves the LLC hit rate of big cores by 2.79% on average, and compared with SRCP, HAPC improves the performance of big cores by 1.18% on average. This shows that the use efficiency of the big core for LLC has been improved, which is an important reason for the performance improvement. In summary, by sensing the memory access characteristics of heterogeneous cores, HAPC can effectively retain the shared cache blocks accessed by big cores, improve the LLC hit rate and IPC of big cores, and ultimately improve the overall performance of asymmetric multi-core systems.

## 5. Discussion

### 5.1. Effect of HAPC on the Parsec Sim_Medium Program

We also evaluated HAPC on the parsec sim_medium benchmarks in gem5 and observed that it provides similar performance to that realized on the sim_small benchmarks. Sim_medium benchmarks are medium-scale experiments, while sim_small benchmarks are small-scale experiments. Figure 7 shows the IPC speedup over the baseline of each replacement policy. HAPC provides the best average performance across all configurations. On average, using the heterogeneous-aware partitioned cache replacement policy results in a 3.16% performance benefit compared to LRU, and a 1.75% performance gain compared to SRCP.

Figure 8 shows the improvement of LLC hit rate over LRU of each replacement policy for parsec sim_medium benchmarks. On average, Figure 8 shows that HAPC achieves an average improvement of LLC hit rate of 3.51%, while SRCP achieves an average improvement of LLC hit rate of 1.61%. These improvements indicate that HAPC is highly modular and shows significant performance benefits on test programs of different scales.

### 5.2. Overheads of HAPC

The RCT in HAPC uses extra space because it stores extra information for monitoring the usage of cache blocks in LLC. Each entry of RCT corresponds to one cache block in LLC. Each entry has 4 bits for LC and 4 bits for SC. A 2 MB LLC has 32 K cache blocks. HPAC imposes 32 KB storage overhead, and about 1.5% of the LLC size. Overall, HAPC offers a practical replacement policy for eliminating intercore interference in the shared LLC on an asymmetric multi-core processor.

### 5.3. Varying Number of Cores

To realistically model a modern commercial asymmetric multi-core processor, we simulate an eight-core system with four big cores and four little cores. Figure 9 shows the performance improvement of big cores in the eight-core system. We make two observations from Figure 9. First, HAPC consistently outperforms LRU and SRCP in the eight-core system; HAPC outperforms LRU and SRCP by 5.86% and 3.60%, respectively. Second, HAPC’s performance improvement over prior replacement policies increases as core count increases.

## 6. Conclusions

With the diversification of computer systems and application requirements, asymmetric multi-core architecture will become the development direction of high-performance and low-power processors. Under the asymmetric multi-core architecture, we propose a heterogeneous-aware partition cache (HAPC) replacement policy. HAPC dynamically adjusts the reuse weight of cache blocks by sensing the difference in memory access characteristics of heterogeneous cores, reducing the interference of little cores on the memory access behavior of big cores, thereby ensuring that cache blocks reused by big cores are not easily evicted in the replacement policy. The experimental evaluation shows that HAPC is superior to the traditional LRU replacement policy and the sharing and reuse-aware SRCP replacement policy. HAPC can better play the high-performance advantages of the big cores under the asymmetric multi-core architecture without affecting the performance of the little cores, improving the use efficiency of LLC and improving the overall performance of the system.

In the context of a multi-core processor, dynamic cache partitioning is also an effective method that has been used to manage shared cache. The replacement policy we proposed focuses on static cache partitioning; however, the same can be implemented for dynamic partitioning. In our future work, we will explore the heterogeneous-aware partition cache replacement policy on top of dynamic partitioning.

## Figures and Tables

**Figure 1 micromachines-13-02014-f001:**
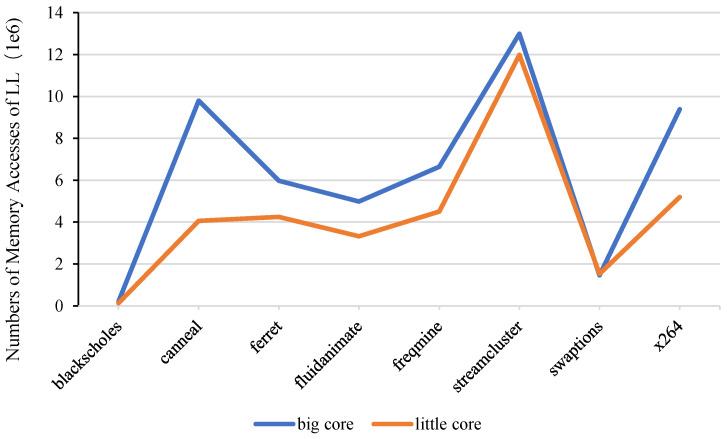
Difference in the number of LLC accesses between big and little cores.

**Figure 2 micromachines-13-02014-f002:**
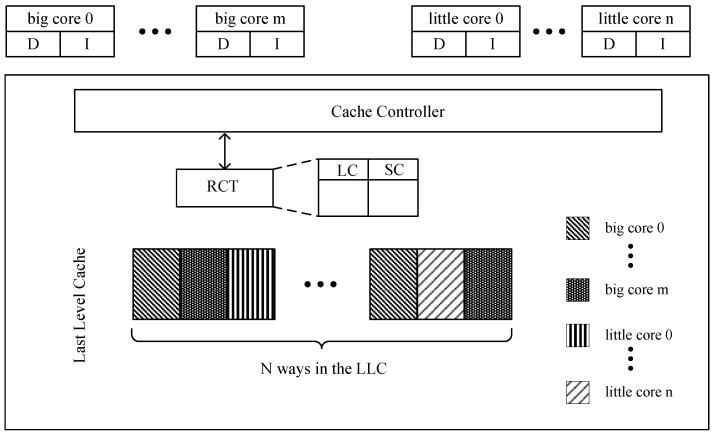
Overview of the HAPC.

**Figure 3 micromachines-13-02014-f003:**
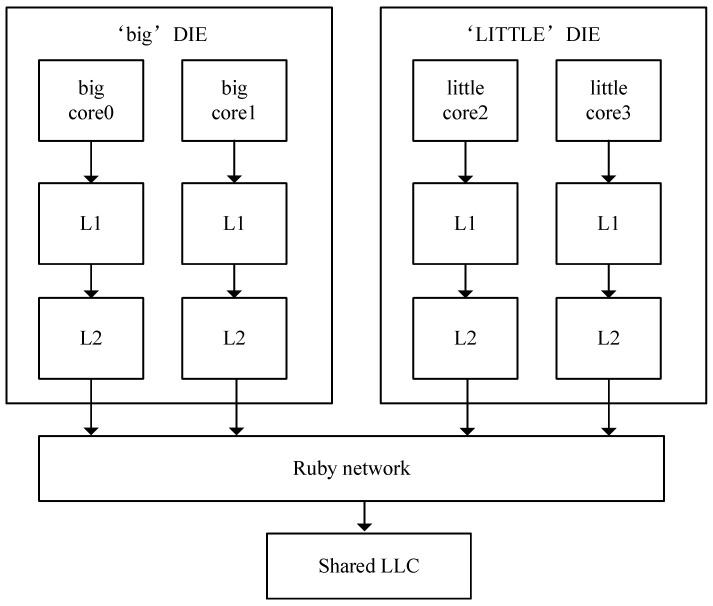
System architecture of experiment.

**Figure 4 micromachines-13-02014-f004:**
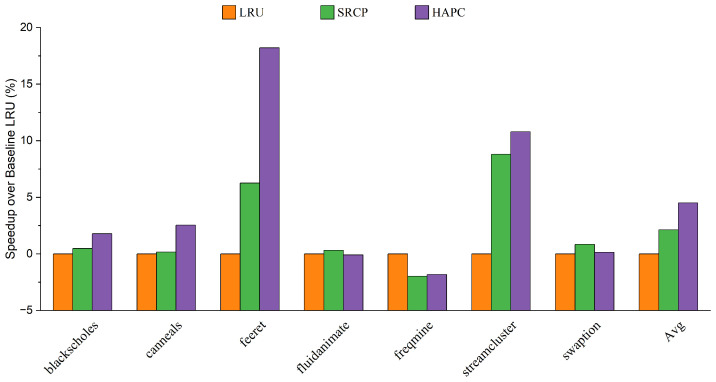
Speedup over baseline LRU of big cores.

**Figure 5 micromachines-13-02014-f005:**
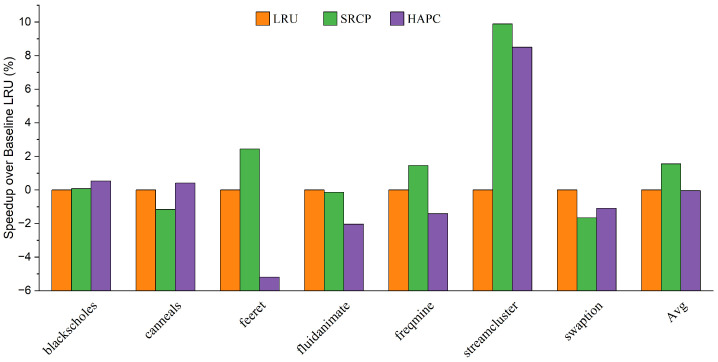
Speedup over baseline LRU of little cores.

**Figure 6 micromachines-13-02014-f006:**
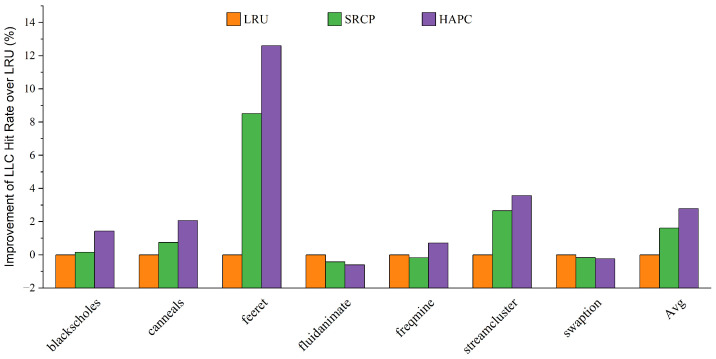
Improvement of hit rate over baseline LRU.

**Figure 7 micromachines-13-02014-f007:**
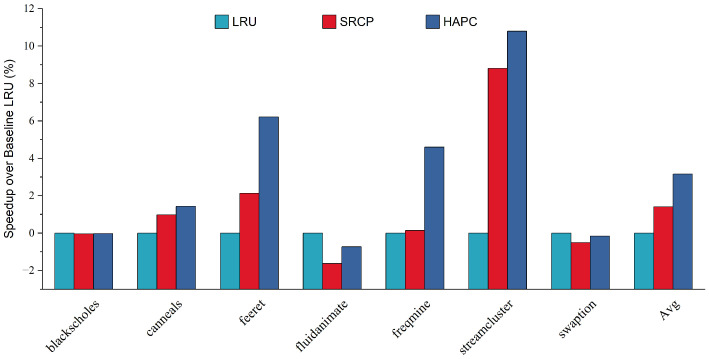
Speedup over baseline LRU.

**Figure 8 micromachines-13-02014-f008:**
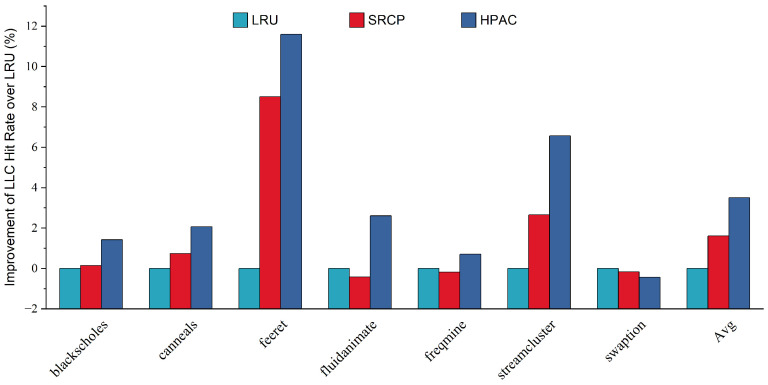
Improvement of hit rate over baseline LRU.

**Figure 9 micromachines-13-02014-f009:**
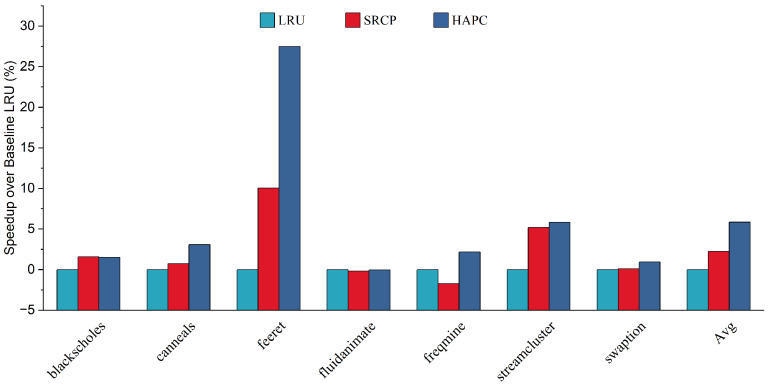
Speedup over baseline LRU.

**Table 1 micromachines-13-02014-t001:** Baseline configuration.

Core	Big Core	Little Core
ISA	ARMv8 (64 bit)	ARMv8 (64 bit)
Frequency	2.0 Hz	1.4 Hz
Pipeline	Out-of-order	Out-of-order
Issue width	6	4
Fetch width	16	4
Pipeline stages	Big core	Little core
L1 cache (I & D)	32 KB/2-way	32 KB/2-way
L2 cache	128 KB/2-way	128 KB/2-way
LLC	1 MB–8 MB/16-way	

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
