# Peer review of "A Heterogeneity-Aware Replacement Policy for the Partitioned Cache on Asymmetric Multi-Core Architectures"

_micromachines, 2022, doi:10.3390/mi13112014_

Round 1

Reviewer 1 Report

The paper proposed a partition-aware replacement policy for LLC in heterogeneous multicore processors. The paper is well-written (except few typos) and easy to understand. However, I have found some issues in the current version:

  1. The hardware overhead on RCT is not discussed in the paper. At least the storage overhead of the proposed technique must be included.

  2. Why the proposed policy is implemented only on top of static partitioning? Static partitions are not practical and they degrade the performance. The authors should use the proposed replacement policy on top of dynamic partitioning as defined in the following paper.

M. K. Qureshi and Y. N. Patt, "Utility-Based Cache Partitioning: A Low-Overhead, High-Performance, Runtime Mechanism to Partition Shared Caches," 2006 39th Annual IEEE/ACM International Symposium on Microarchitecture (MICRO'06), 2006, pp. 423-432, doi: 10.1109/MICRO.2006.49.

  1. Why the performance of light cores is not included?
  2. A modified LRU as mentioned in the above paper can also perform the partition-aware replacement. The authors have applied LRU over the complete LLC set, on top of partitioning. Which is not appropriate for a partitioned cache. Suppose the associativity of the LLC is 16 and the cache is partitioned among two cores as (10, 6). A modified LRU can handle the victim selection such that the partition should remain (10, 6). This modified replacement policy can also handle the dynamic partition change automatically. The authors are advised to use this modified LRU to compare it with the proposed idea.

Author Response

Response to Reviewer 1 Comments

Thank you for your feedback and suggestions. We have incorporated all comments and suggestions into the revised version of the paper. The following is a summary of the revision we conducted.

Point 1: The hardware overhead on RCT is not discussed in the paper. At least the storage overhead of the proposed technique must be included.

Response 1: Thank you very much for raising this issue. The hardware overhead on RCT is 32KB, and about 1.5% of a 2MB LLC.

We have added the analysis of hardware overhead on RCT in section 5.2.

Point 2: Why the proposed policy is implemented only on top of static partitioning? Static partitions are not practical and they degrade the performance. The authors should use the proposed replacement policy on top of dynamic partitioning as defined in the following paper.

  1. K. Qureshi and Y. N. Patt, "Utility-Based Cache Partitioning: A Low-Overhead, High-Performance, Runtime Mechanism to Partition Shared Caches," 2006 39th Annual IEEE/ACM International Symposium on Microarchitecture (MICRO'06), 2006, pp. 423-432, doi: 10.1109/MICRO.2006.49.

Response 2: Thanks for your question. The main goal of HAPC is to analyse the memory access characteristics of big cores and little cores at runtime, and guide the replacement policy by sensing the LLC requirements of heterogeneous core. HAPC focuses on static cache partitioning; however, the same can be implemented for dynamic partitioning. In our future work, we will explore the heterogeneous-aware partition cache replacement policy on top of dynamic partitioning.

In response to the raised question, the discussion of the question is shown in the last paragraph of Section 6.

Point 3: Why the performance of light cores is not included?

Response 3: Thanks for the comment. In response to the raised question, we have added the performance of light cores in Figure. 5. The performance of light cores decreases by only 0.04% on average, and the decline has little effect on the overall performance.

Figure 5. Speedup over baseline LRU of little cores.

Point 4: A modified LRU as mentioned in the above paper can also perform the partition-aware replacement. The authors have applied LRU over the complete LLC set, on top of partitioning. Which is not appropriate for a partitioned cache. Suppose the associativity of the LLC is 16 and the cache is partitioned among two cores as (10, 6). A modified LRU can handle the victim selection such that the partition should remain (10, 6). This modified replacement policy can also handle the dynamic partition change automatically. The authors are advised to use this modified LRU to compare it with the proposed idea.

Response 4: Thank you for raising this question. The LRU as mentioned in this paper has been modified and has implemented the method you mentioned. It only selects eviction blocks in a partition cache belonging to a core, not in the complete LLC set.

Reviewer 2 Report

In this paper, the authors present a cache partitioning mechanism for asymmetric many-core systems. The paper is overall well-written with a clear structure and a nice flow. How ever, there are some unclear points.

- The gains of the proposed method are small for the majority of the experiments. So a more elaborate discussion in needed. Why should someone select your method with such little gains when there are other methods with greater results.

- The authors in [Q1][Q2] showed that priority-aware scheduling can have greater speed up comparing to cache partitioning techniques. In the presented paper, the authors do not investigate co-executing scenarios which is the main case in modern systems. Please provide specific details on such cases and how the proposed method compares with [Q1], [Q2].

[Q1] Kundan, Shivam, and Iraklis Anagnostopoulos. "Priority-aware scheduling under shared-resource contention on chip multicore processors." In 2021 IEEE International Symposium on Circuits and Systems (ISCAS), pp. 1-5. IEEE, 2021.

[Q2] Kundan, Shivam, Theodoros Marinakis, Iraklis Anagnostopoulos, and Dimitri Kagaris. "A Pressure-Aware Policy for Contention Minimization on Multicore Systems." ACM Transactions on Architecture and Code Optimization (TACO) 19, no. 3 (2022): 1-26.

- More configurations in terms of number of cores should be used. Two clusters of two cores is not a representative example of current architectures

- More benchmarks should be used. Parsec is a quite old benchmark suite. 

- There should also be a discussion about the overhead of the presented method and where does it stand compared to the other methods

Author Response

Response to Reviewer 2 Comments

Thank you for your feedback and suggestions. We have incorporated all comments and suggestions into the revised version of the paper. The following is a summary of the revision we conducted.

Point 1: The gains of the proposed method are small for the majority of the experiments. So a more elaborate discussion in needed. Why should someone select your method with such little gains when there are other methods with greater results.

Response 1: Thank you very much for your comments. We propose HAPC to reduce inter-core interference in an asymmetric multi-core architecture. HAPC improves the performance of big cores by 4.57% and 2.44% on average over LRU and SRCP, respectively. Applications like feeret and streamcluster which are inter-thread communication intensive, show significant improvement in performance than LRU and SRCP approaches.  Fluidanimate and swaption are memory non-intensive applications, threads in those applications run in parallel and do not coordinate much, so that HPAC shows negligible performance improvement than LRU. For applications like blackscholes and canneals, it gets benefited with HAPC due to decreased inter-core interference in an asymmetric multi-core architecture.

We use the above analysis in the section 4.2 to explain your concern clearly.

Point 2: The authors in [Q1][Q2] showed that priority-aware scheduling can have greater speed up comparing to cache partitioning techniques. In the presented paper, the authors do not investigate co-executing scenarios which is the main case in modern systems. Please provide specific details on such cases and how the proposed method compares with [Q1], [Q2].

[Q1] Kundan, Shivam, and Iraklis Anagnostopoulos. "Priority-aware scheduling under shared-resource contention on chip multicore processors." In 2021 IEEE International Symposium on Circuits and Systems (ISCAS), pp. 1-5. IEEE, 2021.

[Q2] Kundan, Shivam, Theodoros Marinakis, Iraklis Anagnostopoulos, and Dimitri Kagaris. "A Pressure-Aware Policy for Contention Minimization on Multicore Systems." ACM Transactions on Architecture and Code Optimization (TACO) 19, no. 3 (2022): 1-26.

Response 2: Thanks for your question. The main goal of HAPC is to makes a distinction between big cores and little cores in the cache replacement policy to avoid inter-core interference. Priority-aware scheduling is an important question that has been unstudied to improve the performance of prioritized applications executing under shared-resource contention. Some research find that smart cache replacement reduces the burden on software to provide intelligent scheduling decisions. Herein, these priority-aware scheduling and HAPC combined can together improve performance.

We added specific detail on priority-aware scheduling and co-executing scenarios in Section 2.2.

Point 3: More configurations in terms of number of cores should be used. Two clusters of two cores is not a representative example of current architectures.

Response 3: Thanks for your valuable suggestion. In response to this comment, we have conducted new experiments to report on the performance of the HAPC under different configurations. We have tested HAPC under two clusters of four cores. From the results in Figure. 9, HAPC consistently outperforms LRU and SRCP in the configuration. HAPC outperforms LRU and SRCP by 5.86% and 3.60% respectively.

In response to the raised question, the discussion of the experiments is shown in the Section 5.3.

Figure 9. Speedup over baseline LRU

Point 4: More benchmarks should be used. Parsec is a quite old benchmark suite.

Response 4: Thanks for your question. We evaluate HAPC on the parsec 3.0 benchmark suite released in 2017. The PARSEC benchmark suite, containing 13 applications from six emerging domains, has been widely used for evaluating chip multiprocessors (CMPs) systems since it is released. And in recent years, many researchers developed the parsec 3.0 version, so that the parsec 3.0 covers many emerging domains that are representative for future applications.

Point 5: There should also be a discussion about the overhead of the presented method and where does it stand compared to the other methods.

Response 5: Thank you very much for raising this issue. The hardware overhead on RCT is 32KB, and about 1.5% of a 2MB LLC.

We have added the analysis of hardware overhead on HAPC in section 5.2.
